# Antibacterial Performance of a Mussel-Inspired Polydopamine-Treated Ag/Graphene Nanocomposite Material

**DOI:** 10.3390/ma12203360

**Published:** 2019-10-15

**Authors:** Jianming Liao, Shuaiming He, Shasha Guo, Pengcheng Luan, Lihuan Mo, Jun Li

**Affiliations:** State Key Laboratory of Pulp and Paper Engineering, South China University of Technology, Guangzhou 510641, China; feliaojm_1992@mail.scut.edu.cn (J.L.); he.shuaiming@gmail.com (S.H.); feshasha.guo@mail.scut.edu.cn (S.G.); luanpc1991@gmail.com (P.L.)

**Keywords:** antibacterial, GO, dopamine, Ag, nanocomposite

## Abstract

Graphene-based nanocomposites have attracted tremendous attention in recent years. In this study, a facile yet effective approach was developed to synthesize reduced graphene oxide and an Ag–graphene nanocomposite. The basic strategy involved in the preparation of reduced graphene oxide includes reducing graphene oxide with dopamine, followed by in situ syntheses of the Ag-PDA-reducing graphene oxide (RGO) nanocomposite through adding AgNO_3_ solution and a small amount of dopamine. The nanocomposite was characterized by transmission electron microscopy (TEM), atomic force microscopy (AFM), X-ray diffraction (XRD), FTIR spectra, Raman spectra, ultraviolet-visible (UV-vis), and X-ray photoelectron spectroscopy (XPS), results indicated that a uniform PDA film is formed on the surface of the GO and GO is successfully reduced. Besides, the in situ synthesized Ag nanoparticles (AgNPs) were evenly distributed on the RGO surface. To investigate antibacterial properties Ag-PDA-RGO, different dosages were selected for evaluating the antibacterial activity against Gram-positive bacteria *Staphylococcus aureus* and Gram-negative bacteria *Escherichia coli*. The Ag-PDA-RGO nanocomposites displayed excellent antibacterial property. The antibacterial ratio reached 99.9% against *S. aureus* and 90.9% against *E. coli* when the dosage of 100 mg/L Ag-PDA-RGO nanocomposites was 100 μL. The novel Ag-PDA-RGO nanocomposite prepared by a facile yet effective, environmentally friendly, and low-cost method holds great promise in a wide range of modern biomedical applications.

## 1. Introduction

Microorganisms are omnipresent in our environment. Once the bacteria adhere to the surface of an object, a persistent biofilm will form on the surface. It is usually difficult to kill the bacteria inside because the surface biofilm cannot be affected by common phagocytosis and antibiotics [1]. Thus, it is imperative to develop highly efficient antibacterial materials to address the challenges. Nanocomposites containing two or more nanomaterials have attracted more attention due to their unique functions, such as in antimicrobial, catalytic, electronic, and optical applications [2]. Ag nanoparticles (AgNPs) and graphene nanocomposites have a wide range of applications including antibacterial applications, however, each material has its own shortcomings that need to be addressed [3].

AgNPs and Ag-based composites have attracted extensive attention owing to their broad-spectrum antibacterial properties and potential applications [4], such as biomedical, food production and water purification. The antibacterial mechanism of nano-silver material is not fully understood to date, but it has been proposed that the free silver ions released by AgNPs can disturb the bacterial ATP products and DNA replication [5]. In addition, AgNPs may generate reactive oxygen species (ROS) or directly damage the cell membrane leading to the release of intracellular substances. AgNPs tend to aggregate during the preparation process, resulting in the weakening of their antibacterial properties [6]. Thus, it is important to find an effective support material to enhance the stability of AgNPs.

Owing to the large surface area, graphene is a promising candidate as a support substrate for AgNPs stabilization. It is a monoatomic 2D carbon material with excellent electrical, optical, catalytic, mechanical, and antibacterial properties [7,8,9,10]. In the past few years, various methods have been reported for preparing high-quality graphenes, such as micromechanical exfoliation of graphite [11], chemical vapor deposition (CVD) [12], and chemical reduction of graphene oxide (GO) [13,14,15,16]. Generally, the chemical reduction is a commonly used method for preparing graphene. However, the reagents for reducing graphene oxide (RGO), such as hydrazine, dimethylhydrazine, hydroquinone, and NaBH_4_, are usually toxic or dangerous reagents. In addition, in the absence of dispersants or surfactants, the reduced graphene oxide tends to undergo irreversible agglomeration by π–π stacking interactions [7]. Therefore, a non-toxic and environmentally friendly method for the preparation of stable RGO needs to be developed.

Compared to covalently bonded AgNPs/graphene nanocomposites, non-covalently modified graphene materials have facile and irreversible advantages in the deposition process of AgNPs, and have little change in intrinsic structure and properties of graphene [7]. Therefore, a reductive adhesive material such as polydopamine (PDA) can be used to simultaneously achieve non-covalent modification of graphene and adhesion of AgNPs. Polydopamine, inspired by the unique adhesive foot protein found in mussels, can easily attach to different types of substrates even on the wet surface [17,18]. It is non-toxic and environmentally friendly. Research shows that dopamine can self-polymerize under suitable alkaline condition, and form a polydopamine nano-coating on the surface of inorganic and organic material with controllable thickness and durable stability [19,20,21,22]. There are many functional groups, such as catechol, amine, and imine, existing in the PDA coating, and these groups can provide starting sites for grafting molecular chains and chelate points for metal ions, which can further promote the in-situ synthesis of multifunctional nanocomposites owing to its reducing properties [23,24,25]. It has been demonstrated that polydopamine show excellent biocompatibility and negligible cytotoxicity. These properties allow us to utilize PDA as both a reductant and a surface modifier to prepare AgNPs/graphene nanocomposite. In addition, the reducibility of dopamine enables Ag^+^ to be rapidly reduced and deposited on the graphene surface.

In this study, we develop a facile yet rapid method for synthesizing stable PDA modified RGO and constructing antibacterial nanocomposites (Ag-PDA-RGO), where PDA acts as a hydrophilic and reducing biopolymer to achieve the reduction and surface functionalization of GO. The antibacterial nanocomposites (Ag-PDA-RGO) were prepared by the in-situ reduction of silver ions on the surface of PDA-RGO flakes. The chemical structure, morphology and antibacterial property of the synthesized antibacterial nanocomposites were systematically investigated.

## 2. Materials and Methods

### 2.1. Materials

Silver nitrate (>99.8%), dopamine hydrochloride (>99.9%), graphite powder (325 mesh, 99.9995%), sodium nitrate (>99.0%), and tris (hydroxymethyl) aminomethane were purchased from Sigma-Aldrich (Sigma-Aldrich, Shanghai, China). *E. coli* (ATCC 11229) and *S. aureus* (ATCC 6538) were obtained from Guangdong Microbiology Culture Center (Guangzhou, China). Plate agar (PAC), beef extract, and bacterial peptone were purchased from Guangdong Huankai Microbial Sci. and Tech. Co., Ltd. (Guangzhou, China).

### 2.2. Preparation of GO

GO was prepared by the Hummers’ method [26]. Firstly, graphite powder (3 g) and NaNO_3_ (1.5 g) was added to the concentrated H_2_SO_4_ (69 mL) solution, the mixture solution was then kept at 0 °C. Secondly, KMnO_4_ (9.0 g) was carefully added into the solution and the temperature was kept below 20 °C. After that, the reaction temperature was maintained at 35 °C for 30 min, then an appropriate amount of water (138 mL) was slowly added into the solution, and the reaction temperature was maintained at 98 °C for 15 min. The reaction was allowed to cool for 10 min using a water bath. Lastly, additional water (420 mL) and 30% H_2_O_2_ (3 mL) were added to generate an exothermic reaction, followed by naturally cooling to ambient temperature to obtain the final GO solution.

### 2.3. Preparation of Ag-PDA-RGO Nanocomposites

The schematic diagram of the preparation of Ag-PDA-RGO nanocomposite is shown in Figure 1. GO (50 mg) and dopamine hydrochloride (50 mg) were added to Tris (pH = 8.5) buffer solution (200 mL), and the mixed solution was sonicated for 5 min. Then, the mixed solution was poured to a three-necked flask, and placed in a 60 °C water bath for 24 h. 5 mL of the AgNO_3_ solution (10 mg/mL) was added into the flask after the reaction, and 25 mg of dopamine hydrochloride was added for 30 min to ensure that Ag^+^ was reduced to AgNPs. The Ag-PDA-RGO solution was washed by multiple centrifugations. The centrifuged samples were frozen in a refrigerator at −21 °C for several hours and then placed in a freeze dryer (GOLD-SIM, Los Angeles, USA) for 48 h.

### 2.4. Antibacterial Property Test

The shaking method was applied to evaluate the antibacterial activity of Ag-PDA-RGO against Gram-positive bacteria *S. aureus* and Gram-negative bacteria *E. coli*. Of the bacterial suspension 100 μL was inoculated into a conical flask containing different doses of 100 mg/L Ag-PDA-RGO dispersion (40, 60, 80, 100, and 120 μL). The flask was incubated for 1 h at 37 °C in a shaking incubator (ZHICHENG, Shanghai, China) at 100 rpm. The control samples (GO and PDA-RGO) were prepared by a similar method. 100 μL of the bacterial suspension cultured for 1 h was diluted to 10, 10^2^ and 10^3^ times and 1 mL of each dilution was transferred on a plate. Pour agar with temperature of approximately 45 °C into the plate. The plate was then kept in an incubator (OYSEI, Chongqing, China) at 37 °C for 24 h. The antibacterial ratios were calculated using the following equation:(1)$$\mathrm{Antibacterial}\;\mathrm{ratios}\left(\%\right)=\frac{X_{0}-X_{1}}{X_{0}}\times 100\%,$$
where *X*_0_ and *X*_1_ are the numbers of the colonies incubated with control and Ag-PDA-RGO nanocomposites, respectively.

### 2.5. Characterization

(1) Morphological observation:

TEM and energy dispersive X-ray (EDX) mapping were recorded on the transmission electron microscopy (JEM-2100F, JEOL, Tokyo, Japan) equipped with an energy dispersive X-ray (EDX) spectroscopy (JEOL, Tokyo, Japan), operating at 200 kV. Ultrasonic dispersed GO and PDA-RGO suspension droplets were placed on a 220 mesh micro-grid support films (Antpedia, Guangzhou, China). Excess liquid was absorbed by filter paper and dried at room temperature. The Ag-PDA-RGO sample was ultrasonically dispersed (Sonics, Newtown, CT, USA) and dropped on a mica sheet followed by air drying for atomic force microscopy (AFM) analysis. AFM analysis was performed on a Bruker Multimode 8 microscope (Bruker, Karlsruhe, Germany).

(2) XRD analysis:

Wide-angle XRD patterns were recorded on an X-ray diffractometer (D8 ADVANCE, Bruker, Karlsruhe, Germany) using Cu Kα radiation (40 kV, 40 mA, 10° min^−1^ from 5 to 90°).

(3) UV spectroscopy:

Room temperature UV-vis absorption spectroscopy was conducted on a UV spectrophotometer (UV-2600, SHIMADZU, Kyoto, Japan) using a cuvette.

(4) Raman spectra analysis:

Raman spectra were recorded on a multichannel confocal micro-spectrometer (LabRAM Aramis, HORIBA Jobin Yvon, Paris, France) with 532 nm laser excitation.

(5) FT-IR spectroscopy:

The samples were placed in an agate mortar (Shhk, Shanghai, China) and mixed with KBr (spectral pure) after freeze-dried, the samples were subsequently pressed into a transparent sheet and then measured by a Fourier transform infrared spectrometer (VERTEX 70, Bruker, Karlsruhe, Germany) in the region of 400–4000 cm^−1^.

(6) X-ray photoelectron spectra (XPS) analysis:

XPS were recorded on photoelectron spectrograph (Kratos Axis Ultra DLD, Kratos, Manchester, Britain), using a standard Al Kα X-ray source (75 W) and an analyzer pass energy of 160 eV. Samples were freeze-dried and mounted using the double-sided adhesive tape and binding energies were referenced to the C (1s) binding energy of adventitious carbon contamination taken to be 284.6 eV.

## 3. Results and Discussion

### 3.1. Dopamine Polymerization on the Surface of GO

The mechanism of dopamine self-polymerization remains unclear. Inspired by the adhesive proteins in mussels’ surface, Lee et al. used dopamine to form an adhesive film onto a wide range of inorganic and organic materials [18]. Mussel-inspired PDA has a high concentration of catechol and amine structure. It has an excellent affinity and can be combined with most organic and inorganic material. The TEM images of GO and PDA-RGO were shown in Figure 2a,b. The PDA-RGO was centrifuged and washed against distilled water repeatedly. It can be seen from Figure 2b that a uniform PDA film was formed on the surface of the GO. In addition, since the PDA molecular chain contains many amino groups, the presence of the N element could be detected on the surface by EDX (Figure 2d) analyses. Moreover, the detected C and N element (Figure 2c,d) demonstrated that PDA was evenly distributed on the surface of the GO nanosheets, which could act as an effective binder for GO and AgNPs and improves certain key properties of graphene such as dispersibility and affinity. After loading PDA, the dispersibility of RGO in water was greatly improved. Besides, the abundant active sites on PDA further improved the reactivity of RGO, which is beneficial for the in situ reductions of AgNPs.

Dopamine self-polymerization may involve the oxidation of the catechol to a quinone, which is similar to the way melanin is formed. Thus, the color of GO solution transfers from brown-yellow to deep dark during the dopamine polymerization process. The color transition process can be seen in Figure 3c. In the process of dopamine polymerization, it is accompanied by the reduction of GO due to the oxidation of dopamine. The Raman spectrum of GO and PDA-RGO are shown in Figure 3a. The I_D_/I_G_ value increased from 0.84 to 1.01, indicating a partial sp2 region formed and incomplete reduction of GO after reacting with dopamine. Due to the limitations of chemically reduced graphene oxide, the reduction of GO may be incomplete, resulting in a broad peak in the Raman spectrum. This incomplete reduction of GO may have a beneficial effect on the surface of GO, which has better hydrophilicity and colloidal stability than fully reduced GO, while fully reduced RGO is poor due to its hydrophobicity [27]. This is conducive to the dispersion of Ag-PDA-RGO nanocomposites. Similar results could also be obtained from the UV spectrum and XRD patterns of GO, PDA-RGO. The absorption peak of GO at 230 nm had a certain degree of redshift in the PDA-RGO ultraviolet curve. As shown in the XRD patterns (Figure 3b), the sharp characteristic peak (2θ = 10.7°) of GO disappeared after reacting with dopamine. Besides, a new broad diffraction peak (2θ = 23.4°) appeared, which was similar to the typical diffraction peak of graphite, indicating that GO was reduced.

### 3.2. AgNPs Deposition on the Surface of PDA-RGO

In addition to its adhesive ability, another valuable property of PDA is its chemical structure that incorporates many functional groups such as catechol, amine, and imine [19]. These functional groups can provide binding sites for Ag^+^ and reduce it to AgNPs [28]. At the same time, the catechol and the hydroxyl groups of PDA are oxidized to quinone and carbonyl groups, respectively. The changes in the functional groups were confirmed by FTIR. The infrared spectra of GO, PDA-RGO, and Ag-PDA-RGO are shown in Figure 3d. The curve of GO had an absorption peak at 1721 cm^−1^, which represents a stretching vibration peak of C=O in the carboxyl group. The peak of the carboxyl group disappeared in the curves of PDA-RGO and Ag-PDA-RGO, indicating that the GO was reduced after the reaction with dopamine. The peaks at 1462 cm^−1^ and 1230 cm^−1^ represent the bending vibration of –N–H and the –C–N stretching vibration peak, respectively, indicating that the original epoxy group reacted with dopamine [29,30]. Therefore, FTIR results indicate the existences of PDA film on the surface of GO, which can help provide a large number of binding sites for Ag^+^, such as catechol, amine, and imine.

AgNPs have been demonstrated to be an ideal material that can be applied in surface-enhanced Raman scattering (SERS) [31]. As shown in Figure 3a, graphene had two distinct peaks (D and G peaks) at ~1350 cm^−1^ and ~1610 cm^−1^, respectively. In the curve of Ag-PDA-RGO, the graphene Raman scattering signal was obviously enhanced after the deposition of AgNPs, while the curve signals of GO and PDA-RGO were relatively weak. This might be derived from the surface-enhanced Raman scattering of Ag in the surface of PDA-RGO [32].

AgNPs and graphene have special UV absorption peaks [14,33]. The UV spectra of GO, PDA-RGO, and Ag-PDA-RGO are shown in Figure 3e. There was an absorption peak at 230 nm, which is the characteristic absorption peak of GO. However, this peak in the Ag-PDA-RGO nanocomposite had a certain degree of redshift in the ultraviolet curve, indicating that a certain degree of reduction occurred after the reaction of GO with dopamine and the PDA film formed can help tightly anchor and evenly distribute the AgNPs on the surface. A new absorption peak appeared at 451 nm, which indicates Ag^+^ was reduced to AgNPs.

Furthermore, the X-ray diffraction (XRD) patterns of GO, PDA-RGO, and Ag-PDA-RGO are shown in Figure 3b. The Ag-PDA-RGO shows sharp peaks at 2θ = 38.1°, 44.2°, 64.4°, and 77.3°, which could be indexed to the (1 1 1), (2 0 0), (2 2 0), and (3 1 1) of the face-centered cubic crystal silver structure, respectively [34,35]. It demonstrated that there exists metallic silver on the surface of PDA-GO flakes. This result also could be proved by the AFM and TEM image (Figure 3f and Appendix A). AFM results indicate that the surface of Ag-PDA-RGO was rough, and a large number of nanoparticles were evenly distributed on the RGO surface. These nanoparticles were AgNPs.

X-ray photoelectron spectroscopy (XPS) was used to further verify the chemical state of Ag and the reaction between dopamine and GO (Figure 4). The C 1s XPS spectra of Ag-PDA-RGO nanocomposite (Figure 4b) could be assigned to five components, which the binding energies at 284.6, 285.5, 286.4, 287.8, and 288.9 eV represent C-C, C-N, C-O, C=O, and O-C=O species, respectively. It should be noted that the peak component of C–N species at 285.5 eV was attributed to the amine group of the dopamine. As we can see from the Figure 4c, the N 1s core-level spectrum of the Ag-PDA-RGO nanocomposite had one peak component with the binding energy at about 400 eV, due to the self-polymerization process of dopamine in the GO surface. Figure 4a,d shows an XPS wide scan and Ag 3d core-level spectra of the Ag-PDA-RGO nanocomposite respectively. The strong signal at a BE of about 370 eV in Figure 4d demonstrated that the existence of Ag element. The Ag (3d) peak is a double peak generated by spin-orbit coupling (3d_5/2_ and 3d_3/2_). The binding energies at 368.1 eV and 374.1 eV were attributed to the Ag 3d_5/2_ and Ag 3d_3/2_ peak [1], respectively, which proved that silver was only present in the metallic state.

### 3.3. Antibacterial Property of Ag-PDA-RGO Nanocomposites

The antibacterial mechanism of AgNPs remains to be understood. AgNPs may adhere to the surface of cell membrane, interfering with cell permeability and its respiratory function [36]. For graphene, it has been demonstrated that its antibacterial activity is attributed to membrane stress caused by sharp edges of graphene nanosheets, which may disrupt cell membrane and leak RNA [10]. Therefore, the antibacterial properties of Ag-PDA-RGO may be caused by the synergistic effects of AgNPs and graphene.

Gram-positive bacteria *S. aureus* ATCC 6538 and Gram-negative bacteria *E. coli* ATCC 11229 were used for antibacterial testing. The antibacterial properties of Ag-PDA-RGO nanocomposites were evaluated by calculating the antibacterial ratios based on the number of bacterial colonies cultured at 37 °C for 24 h with different dosages of Ag-PDA-RGO nanocomposites. The antibacterial ratios of Ag-PDA-RGO nanocomposites against *S. aureus* and *E. coli* are shown in Figure 5a. The results indicate that the antibacterial ratio of Ag-PDA-RGO nanocomposites increased with the dosage increasing. The antibacterial ratio reached 99.9% against *S. aureus* and 90.9% against *E. coli* when the dosage of Ag-PDA-RGO nanocomposites was 100 μL. Figure 5c–g show the digital images of bacterial colonies formed by *S. aureus* and *E. coli* before and after adding different amount of Ag-PDA-RGO nanocomposites, results indicated that the bacteria can not grow with increasing dosage. Figure 5b shows inhibition ratio of *E. coli* after adding 100 μL of 100 mg/L GO, PDA-RGO, and Ag-PDA-RGO solutions. Compared with GO and PDA-RGO, the antibacterial capability of Ag-PDA-RGO nanocomposites was significantly improved.

## 4. Conclusions

Inspired by mussels, we successfully achieved the reduction and surface functionalization of GO. Ag-PDA-RGO nanocomposites were rapidly prepared by in situ deposition of silver nanoparticles on PDA-RGO surface. The results indicate that PDA coatings could act as a binder and improve the surface adhesivity and stability of GO. A uniform PDA film was formed on the surface of GO flakes and the GO was successfully reduced. Besides, the in situ synthesized Ag NPs were evenly distributed on the RGO surface. Compared to previous method, the method greatly shortened the reduction time of Ag^+^ and improved the preparation efficiency of Ag-PDA-RGO nanocomposites. The prepared Ag-PDA-RGO nanocomposites exhibited excellent antibacterial property against Gram-positive bacteria *S. aureus* and Gram-negative bacteria *E. coli*. The novel Ag-PDA-RGO nanocomposite prepared by a facile yet rapid method holds great promise in a wide range of modern biomedical applications.

## Figures and Tables

**Figure 1 materials-12-03360-f001:**
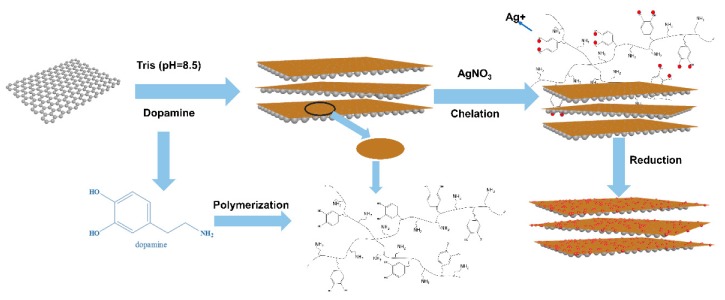
Schematic diagram of dopamine polymerization and reduction of GO and in situ deposition of Ag nanoparticles (AgNPs) on the surface of PDA-reducing graphene oxide (RGO).

**Figure 2 materials-12-03360-f002:**
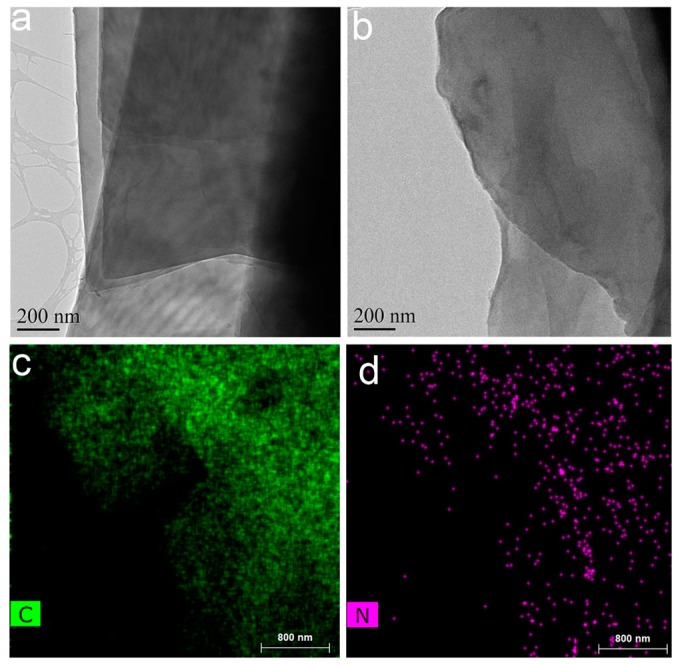
The TEM images of GO (**a**) and PDA-RGO (**b**), and elemental mapping of PDA-RGO (**c**,**d**).

**Figure 3 materials-12-03360-f003:**
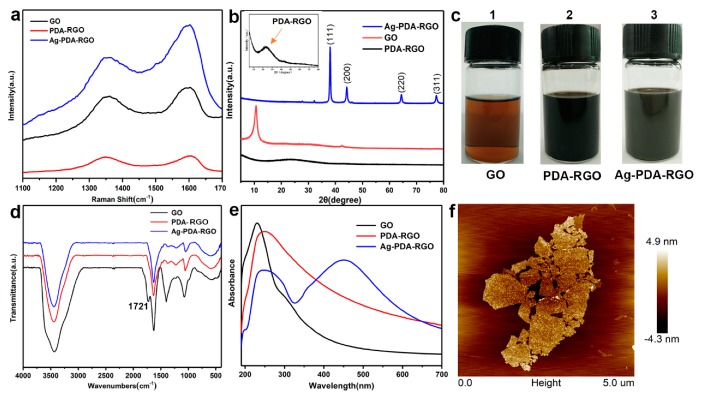
Raman spectra (**a**), XRD pattern (**b**) of GO, PDA-RGO, and Ag-PDA-RGO, and the optical picture (**c**) of GO (1), PDA-RGO (2), and Ag-PDA-RGO (3) aqueous solution. FTIR spectra (**d**), UV–visible spectra (**e**) of GO, PDA-RGO, and Ag-PDA-RGO. Atomic force microscopy (AFM) image (**f**) of Ag-PDA-RGO.

**Figure 4 materials-12-03360-f004:**
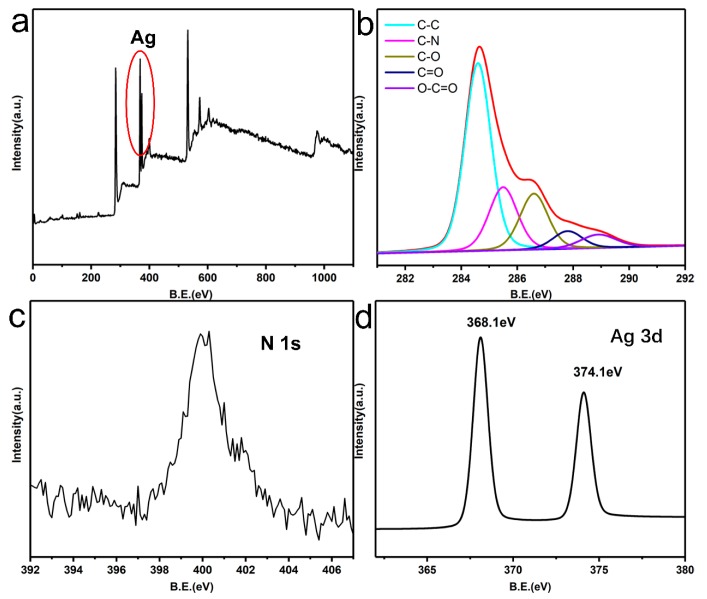
XPS survey scans of Ag-PDA-RGO (**a**), C 1s XPS spectra of Ag-PDA-RGO (**b**), N 1s XPS spectra of Ag-PDA-RGO (**c**), and Ag 3d core-level spectrum of Ag-PDA-RGO (**d**).

**Figure 5 materials-12-03360-f005:**
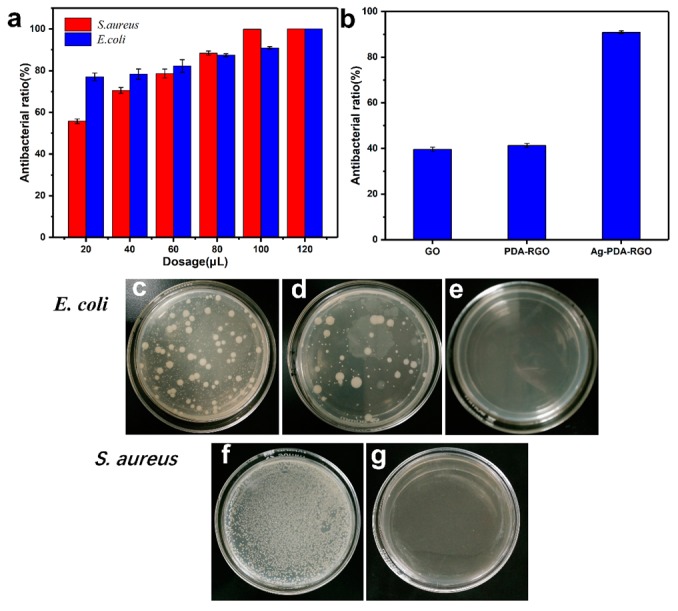
The antibacterial property against *Staphylococcus aureus* and *Escherichia coli* with a different dosage of Ag-PDA-RGO (**a**), antibacterial property comparison against *E. coli* with 100 μL of 100 mg/L GO, PDA-RGO, and Ag-PDA-RGO (**b**), the photographs of initial bacterial colonies formed by *S. aureus* (**c**), *E. coli* (**f**), and the bacterial colonies images after adding 100 μL (**d**,**g**) and 120 μL (**e**) of 100 mg/L Ag-PDA-RGO nanocomposites.

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
