# Peer review of "Antibacterial Performance of a Mussel-Inspired Polydopamine-Treated Ag/Graphene Nanocomposite Material"

_materials, 2019, doi:10.3390/ma12203360_

Round 1

Reviewer 1 Report

The manuscript "A Mussel-inspired Polydopamine-modified Graphene Oxide Based Antibacterial Nanocomposite Material" describe the synthesis method of novel Ag-PDA-RGO nanocomposite and show its antibacterial properties. The manuscript is suggested for publication, after following major concern is addressed:

The major weakness of this study is the absence of antibacterial studies of AgNP, GO and PDA alone. In absence of these groups, it is impossible to fully understand the antibacterial properties. It is important to know if the antibacterial effect of nanocomposite is stronger or weaker than materials used for the synthesis. Authors should explain why Escherichia coli has been chosen for the experiments and describe it in the manuscript. Source of the Escherichia coli should be given Additional bacterial species and antibacterial experiment should be considered such as biofilm analysis (which is described at the beginning of the introduction). Introduction must be revised by adding more information about antibacterial nanocomposites and a better description about the reason to investigate this topic. Escherichia coli should be italicized Results of antibacterial studies should be properly discussed

Author Response

Dear  reviewer,

Thank you very much for your valuable comments and suggestions on our manuscript. We have modified the manuscript or given explanations accordingly. The modified parts in the revised manuscript has been marked with red color. Please see the attachment.

Reviewer 2 Report

Antibacterial properties of Ag and graphene-based nanocomposites is widely investigated which limits the novelty of the presented research (however the design and synthesis proposal looks interesting). Unfortunately, I cannot say that the conclusion concerning the suitability of the nanomaterial for future medical purposes can be stated based on the performed test on antibacterial activity, because:

the test was carried out only on one model bacteria (E. coli). No test for Gram- positive microoganism was proposed (the sensivity of particular groups of bacteria may significantly differ), so at least tests on several bacterial representatives (both Gram-positive and Gram negative) are required the dosage of nanocomponent in antibacterial test was not given properly - Authors presented the volume of the nanocomposite-containing solution without any precise data concerning the REAL concentration of the active nanocomponent in the solution, so it is impossible to refer the antibacterial properties to the certain dose of the applied substance the information concerning the procedure cultivation of bacteria is not sufficient - was it the pour plate method or spread plate method? What agar medium was applied - nutrient agar or another agar medium? 

Author Response

(The authors gave the same response as above.)

Reviewer 3 Report

In a manuscript titled as “A mussel-inspired polydopamine-modified graphene oxide based antibacterial nanocomposite material”, Liao et al. reported on bio-inspired synthesis of AgNPs on reduced graphene oxide using dopamine (Ag-PDA-RGO) and antibacterial ability of Ag-PDA-RGO. Although the Ag-PDA-RGO synthesis method is emphasized in this study, this method is not different from the previously reported method. The manuscript could be accepted after major revisions. Therefore, I recommend rejection of it for Materials.

Escherichia coli should be in italics. In line 21 on page 1, “Ag-PDA-GO” should be changed to “Ag-PDA-RGO”. In line 57 on page 2, polydopamine should be lowercase. In materials and methods, additional information should be provided for companies of used reagents and analytical instruments. For example, Fourier transform infrared spectrometer (VERTEX 70, Bruker, Germany). Authors described that RGO is made by mixing GO and dopamine and then additional dopamine is added to make AgNPs on the RGD surface. It is necessary to explain why additional dopamine should be added. It is thought that excess dopamine may have a rather adverse effect on the generation of AgNPs on the RGO surface. The generation of AgNPs on graphene is considered to be very important in this study. It is necessary to explain how the preparation methods of Ag-PDA-RGO presented in reference 23 and this study are different. In line 159 on page 4, “reduction of dopamine” should be changed to “oxidation of dopamine”. In 3.1 dopamine polymerization on the surface of GO, authors need to present the figure number for the result description and reorganize the figure order. By changing the order of Figure 3a, Figure 3b, Figure 3c, and Figure 3d, authors may create a smooth flow of information. It is also likely that readers understand more easy if the results are explained with the changed order of the figures. In order to show more clearly XRD pattern of PDA-RGO, it is necessary to enlarge XRD data of PDA-RGO. By changing the order of Figure 4a, Figure 4b, Figure 4c, and Figure 4d, authors may create a smooth flow of information. To more strongly demonstrate the generation of AgNPs on surface of PDA-RGO, TEM analysis of Ag-PDA-RGO should be performed. 12. I found some difficulties in reading due to the lack of proficiency in English and some minor errors For example, “As shown in Figure 4 b and c. The C 1s core-level 213 spectrum of Ag-PDA-RGO nanocomposite could be curve-fitted with five peak components, which 214 binding energies at 284.6, 285.5, 286.4, 287.8 and 288.9 eV, representing C-C, C-N, C-O, C=O, and O-215 C=O species respectively.”

Author Response

(The authors gave the same response as above.)

Reviewer 4 Report

The manuscript entitled "A Mussel-inspired Polydopamine-modified Graphene Oxide Based Antibacterial Nanocomposite Material" (Ref: materials-589542) reports the preparation of a novel antibacterial nanocomposite by reduction of silver ions on polydopamine-reduced graphene oxide surfaces.

My overall impression is positive and the manuscript does deserve publication after deep revision.

In general, authors should improve the english grammar of the manuscript. There are several mistakes throughout the text. They must also do a careful revision related to references and the name of the figures. The most important corrections that authors should carry out are:

Line 49: Include the acronym “RGO” in brackets, since it is used afterwards. Line 52: Reference 15 does not correspond to explanation. Select a proper one. Line 76: In the Materials part, give more information about graphite powder (purity, number of mesh). Include the correct name of dopamine (dopamine hydrochloride). Line 80: Method related to reference 12 is not the Hummer’s method, it is a modified one. Report another reference. Line 129: Rewrite the sentence. Line 157: Change “polymerized” by “polymerization”. Line 158: Figure 2 h does not exist. Line 159: Reference to figure 2 b is wrong, that should correspond to figure 3 b. Line 169: Broad peak at 23.4° is not clearly observed. Please, modify y-axis. Line 179: Change “spectrums” by “spectra” Line 180: Point out the peak at 1721 cm-1 in the graph. Line 188: Correct the sentence: “...an ideal material that can be..”. Line 194: Change “spectrums” by “spectra”. Line 199: Change “indicate” by “which indicates”. Line 209: Figure caption is completely wrong and confuse. Line 213: Sentence “As shown in Figure 4 b and c” is a continuation of previous phrase. Line 214: Rewrite the sentence “The C 1s core-level spectrum...” in understandable form. Line 222: Add a reference about binding energies of silver. Line 228: Antibacterial assays of GO and PDA-RGO should be included in the graph for comparison. Furthermore, antibacterial capability of GO is commented in the text. At least, it must be referenced. Line 41: Number of figure is wrong, that corresponds to figure 5.

Author Response

(The authors gave the same response as above.)

Round 2

Reviewer 1 Report

Authors have addressed all of the comments.

Author Response

Dear reviewer,

Thank you for your suggestions.

Reviewer 2 Report

The manuscript has been improved properly

Author Response

Dear reviewer,

Thank you for your suggestion.

Reviewer 3 Report

While the authors have provided responses to my comments, there are several of the issues I raised regarding significance for the approach. I found some minor errors.

The nanocomposite was characterized by transmission electron microscopy (TEM), atomic force microscopy (AFM), X-ray diffraction (XRD), FTIR spectra, Raman spectra, ultraviolet-visible (UV-vis), X-ray photoelectron spectroscopy (XPS). These results indicated that a uniform PDA film is formed on the surface of the GO and GO is successfully reduced. In line 23 on page 1, “S. aureus” should be changed to “Staphylococcus aureus”. In line 168 on page 4, “Figure 2f” should be changed to “Figure 3f”. In line 103 on page 3, “Ag_NPs” should be chaged to “AgNPs”. In line 117 on page 3, “37º-C” should be changed to “37ºC”. Authors descirbed reduction of GO and self-polymerization of dopamine on the GO surface in 3.1 dopamine polymerization on the GO surface through Figure 3f, Figure 3b, and Figure 3d. Authors should reconstitute the order of Figure 3 to create a smooth flow of information. In addition, authors should mark Figure 3d in line 178-179 on page 4. Authors claimed rapid generation of AgNPs on PDA-RGO compared with reference 23 (the reaction of AgNO3 and PDA-RGO for 4 h). Authors should present reaction time of AgNO3 and PDA-RGO under an addition of dopamine hydrochloride in 2.3 preparation of Ag-PDA-RGO nanocomposites. In addition, authors should present differences (originality/novelty) of the approach  with previous methods for Ag-PDA-RGO in introduction, results, and conclusion. Figure 5 c-g show the digital images of bacterial colonies formed by S. aureus and E. coli before and after adding different amount of Ag-PDA-RGO nanocomposites. These results indicated that the bacteria can't grow with increasing dosage. It is demonstrated that the antibacterial effect of AgNPs on Gram-negative bacteria was stronger than Gram-positive bacteria. This is due to the existing difference in cell wall thickness between Gram-positive bacteria (about 30 nm) and Gram-negative bacteria (3-4 nm). As shown in Figure 5a, except for 100 μL of 100 mg/L Ag-PDA-RGO, different dosages of Ag-PDA-RGO followed the previously reported phenomenon. I wonder the reason why antibacterial activity of Ag-PDA-RGO on S. aureus is stronger than E. coli when using 100 μL of 100 mg/L Ag-PDA-RGO.

Author Response

Dear reviewer,

Thank you very much for your valuable comments and suggestions on our manuscript. The modifications in revised-manuscript has been marked with RED color. Please see the attachment.

Reviewer 4 Report

Dear authors, I really appreciate you have fully addressed my concerns, and the manuscript is now satisfactory for publication in this journal. Anyway, I have detected a mistake at line 168; explanation corresponds to Figure 3f, please, correct it.

Author Response

Dear reviewer,

Thank you for your suggestions. The mistake has been corrected. Please see the attachment.
